# *ANK2* Hypermethylation in Canine Mammary Tumors and Human Breast Cancer

**DOI:** 10.3390/ijms21228697

**Published:** 2020-11-18

**Authors:** Johannes J. Schabort, A-Reum Nam, Kang-Hoon Lee, Seok Won Kim, Jeong Eon Lee, Je-Yoel Cho

**Affiliations:** 1Department of Biochemistry, BK21 PLUS Program for Creative Veterinary Science Research and Research Institute for Veterinary Science, College of Veterinary Medicine, Seoul National University, 08826 Seoul, Korea; jan2schabort@snu.ac.kr (J.J.S.); arbjlvz@snu.ac.kr (A.-R.N.); khlee02@snu.ac.kr (K.-H.L.); 2Division of Breast Surgery, Department of Surgery, Samsung Medical Center, Sungkyunkwan University School of Medicine, 81 Irwon-ro, Gangnam-gu, Seoul 06351, Korea; seokwon1.kim@samsung.com (S.W.K.); jeongeon.lee@samsung.com (J.E.L.)

**Keywords:** CMT, hypermethylation, biomarker, cfDNA, HBC

## Abstract

Canine mammary tumors (CMT) constitute the most common tumor types found in female dogs. Understanding this cancer through extensive research is important not only for clinical veterinary applications, but also in the scope of comparative oncology. The use of DNA methylation as a biomarker has been noted for numerous cancers in the form of both tissue and liquid biopsies, yet the study of methylation in CMT has been limited. By analyzing our canine methyl-binding domain sequencing (MBD-seq) data, we identified intron regions of canine *ANK2* and *EPAS1* as differentially methylated regions (DMGs) in CMT. Subsequently, we established quantitative methylation specific PCR (qMSP) of *ANK2* and *EPAS1* to validate the target hypermethylation in CMT tissue, as well as cell free DNA (cfDNA) from CMT plasma. Both *ANK2* and *EPAS1* were hypermethylated in CMT and highlighted as potential tissue biomarkers in CMT. *ANK2* additionally showed significant hypermethylation in the plasma cfDNA of CMT, indicating that it could be a potential liquid biopsy biomarker as well. A similar trend towards hypermethylation was indicated in HBC at a specific CpG of the *ANK2* target on the orthologous human region, which validates the comparative approach using aberrant methylation in CMT.

## 1. Introduction

Canine mammary tumors (CMT) are the most common neoplasia diagnosed in female dogs and are found to be malignant in approximately 50% of cases [1,2,3,4]. Apart from the clear veterinary benefit to understanding the nature of this cancer more clearly, studying CMT can also be beneficial in the field of comparative oncology [5,6,7].

Dogs have been highlighted as excellent animal models for human cancer [5,6], and this includes the use of CMT as a comparative model for human breast cancer (HBC) [7,8,9,10]. The expedited study rate due to faster disease progression and shorter lifespan, histological and mechanistic similarities [5], spontaneous disease occurrence, and exposure to similar environments as humans make dogs ideal models for comparative oncology [11,12]. The similar environments to humans that dogs often inhabit as companion animals are particularly interesting from an epigenetic standpoint, as environmental stimuli have been known to influence epigenetics [13,14]. 

DNA methylation is an important method of epigenetic regulation and thus has often been studied in relation to cancer [15]. Aberrant global hypomethylation and localized, specific hypermethylation of tumor suppressors and hypomethylation of oncogenes are observed in cancer [16]. In CMT, aberrant methylation has been reported for a few genes such as *ERα* [17], *DAPK1*, *MGMT* [18], and *BRCA1* [19], yet gene-specific methylation of CMT remains largely unknown. These methylation changes are often observed early during carcinogenesis [20] making them ideal biomarkers for early detection and prognosis. Tissue biomarkers remain the standard for cancer diagnosis and prognosis, yet in recent years the interest in liquid biopsy biomarkers have risen [21]. 

Liquid biopsies involve the investigation of cell-free DNA (cfDNA) found in the blood, plasma, serum, or other liquids from the body and often contain circulating tumor DNA. They are of clinical importance as they are less invasive and also allow for a more comprehensive view of the cancer biology as opposed to the view offered by tissue biopsies that is limited to a single site of the tumor at a single moment in the progression of the cancer [21]. Methylation of circulating cfDNA has been highlighted as a human cancer biomarker in liquid biopsies [22,23,24]. Previously, our lab has presented the cfDNA hypomethylation of LINE1 as a candidate liquid biopsy biomarker in both CMT and HBC, however, the method of restriction enzyme digestion followed by quantitative-PCR needs to be improved further for clinical use [25].

This study set out to identify and investigate differentially methylated regions (DMRs) in CMT, initially validating our findings in CMT tissues and then also in cfDNA isolated from plasma obtained from dogs with CMT with the intent of establishing candidate tissue and liquid biomarkers for cfDNA. Additionally, we correlated our differentially methylated target regions in our CMT data with HBC data to see if the differential methylation seen in these diseases is consistent across species at the targeted sites.

## 2. Results

### 2.1. Identification of Differentially Methylated Regions

Target hypermethylated DMRs were identified using MBD-seq and corresponding RNA-seq according to the scheme in Figure 1A. From our previous MBD-seq data obtained for 11 pairs of CMT and adjacent normal tissue from NCBI BioProject database (accession number PRJNA601533) [26], 16,061 differentially methylated genes (DMGs) were identified based on the presence of a differentially methylated region (DMR) within the gene body. To develop DNA methylation-based biomarkers applicable to CMT and HBC, we decided to narrow the scope to regions that contained a hypermethylated CpG island in CMT. A separate study could be performed to analyze the hypomethylated cohort. We further narrowed down the 1269 hypermethylated DMGs, based on the fold change (log2 fold change > 1.5). Of these 20 identified targets, *EPAS1*, *ANK2*, *DST* and *RUSC2* indicated a downregulation of gene expression in CMT noted from the RNA-seq data. Of these, *ANK2* and *EPAS1* were at a significant level (*p*-value < 0.05) and were thus selected for further analysis (Figure 1B). These identified CMT DMRs were found in CpG islands in the 21st and 1st introns of *ANK2* and *EPAS1*, respectively, and the increase in methylation in the cancer samples are shown via linear mixed model (LMM) with thresholds of both 10% and 5% (Figure 1C). The overall methylation level detected via the MBD-seq for both target regions were shown to be significantly more methylated in the CMT samples as opposed to the normal samples (Figure 1D). Of the 11 CMT/normal tissue pairs that were used for MBD-seq, 10 were also used in the RNA-seq analysis. These 10 pairs that had data for both MBD-seq and RNA-seq showed a general reverse correlation trend between expression and methylation for both targets, albeit not at a significant level (Figure 1E,F). Yet, the correlation plots and the associated density plots do clearly reveal that for both targets the level of methylation is higher in CMT than in normal, whereas the level of expression is lower. Compared to the paired normal samples an increase in methylation of *ANK2* and *EPAS1* along with a decrease in expression is indicated in 5/10 and 8/10 CMT samples, respectively (Figure 1E,F). Based on the accumulated bioinformatic data, comprising MBD-seq and RNA-seq data, *ANK2* and *EPAS1* were both highlighted as potential CMT biomarkers with DMRs in CMT and were chosen as the targets for the remainder of this study.

### 2.2. Evaluation of Differentially Methylated Regions in CMT and Adjacent Normal Tissue

The target intron regions of *ANK2* and *EPAS1* that were identified through MBD-seq were further analyzed to assess their differential methylation. Quantitative methylation specific PCR (qMSP) was chosen as the method by which methylation was to be analyzed, mainly due to its ability to quantify the methylation of numerous samples simultaneously without the need for exhaustive cloning and sequencing procedures. qMSP is furthermore a very sensitive technique that is able to distinguish a very small amount of methylated CpGs from unmethylated CpGs [27]. However, sequencing was still performed on three randomly selected pairs of CMT and normal gDNA for each target to establish a representative methylation pattern for each target and thus indicate in which areas to design the MSP primers (Figure 2). Both the *ANK2* and *EPAS1* target regions were methylated more in the CMT than the paired normal samples, although the amount of differential methylation varied across the sample pairs (Figure 2).

qMSP was then performed to investigate more sample pairs. The Methylation Index is presented and is based on the demethylation index first introduced by Akirav et al. [28], with the adjustment of measuring the amount of methylated DNA as opposed to unmethylated DNA. This method uses bisulfite sequencing PCR (BSP) primers that flank the MSP region of interest to normalize the MSP readings (Figure 3A). Of the 15 sample pairs analyzed for the *ANK2* target, 12 were more methylated in CMT compared to normal based on the methylation index, whereas nine out of 15 *EPAS1* target samples were more methylated in CMT than in paired normal (Figure 3B). The overall methylation was shown to be significantly more methylated in CMT for both the *ANK2* and *EPAS1* targets, based on paired t-tests (Figure 3C). Receiver operating characteristic (ROC) curves were constructed for both targets to access the sensitivity and specificity of using CMT hypermethylation as biomarkers. *ANK2* had an area under the curve (AUC) of 0.764 and *EPAS1* had an AUC of 0.733 (Figure 3D). Overall, the quantitative MSP results validated what was shown in the MBD and sequencing data; the targeted *ANK2* intron 21 and *EPAS1* intron 1 regions are hypermethylated in CMT and are candidate tissue biomarkers for this disease.

### 2.3. Detection of Differential Methylation in Canine Plasma cfDNA

qMSP has been utilized in previous studies to detect methylation of cfDNA [29,30]. One study found a significant agreement between *ESR1* methylation of paired plasma and primary ovarian cancer tumors [31]. To evaluate whether the hypermethylation trends noted in CMT tissue could also be detected in cfDNA and thus serve as potential liquid biopsy biomarkers for CMT, we conducted the same qMSP procedure on cfDNA isolated from plasma samples of CMT and normal female dogs. The *ANK2* target was analyzed in pooled cfDNA samples from 19 CMT dogs and 10 normal dogs. The hypermethylation trend of the *ANK2* target region in CMT extended to cfDNA as is shown by the significantly higher methylation index (Figure 4A). Interestingly, the *EPAS1* target, which was investigated in cfDNA samples from 10 CMT and 10 normal dogs, did not indicate an increased level of methylation for CMT as was seen in the tissue samples (Figure 4B). In contrast, the CMT cfDNA showed less methylation in CMT than normal plasma for *EPAS1*. Overall, in canine cfDNA, *ANK2* demonstrated significant hypermethylation in CMT samples, which proposes it as a potential liquid biopsy biomarker for CMT.

### 2.4. Orthologous Human Regions Analyzed from TCGA Data

To investigate whether the hypermethylation trend seen at the target regions in CMT would correspond to what is seen in HBC, we mapped the target regions found on the dog genome (CanFam3.1) to the human genome (HG19). This indicated that our 500 bp regions identified from the canine MBD-seq for the *ANK2* and *EPAS1* targets had orthologous human regions of 405 bp with 79% sequence identity and 221 bp with 76% sequence identity, respectively (Appendix A). The *EPAS1* target had an orthologous region that was also located in a CpG island at the 3′ end of the first intron, just as in canines. The *ANK2* target, which is situated in the 21st intron of the canine gene, similarly had an orthologous region in the 21st intron of human *ANK2*, however, this region does not contain a CpG island on the human genome, which makes it unsuitable for study with the qMSP method that we employed in this study. Even so, we investigated data from The Cancer Genome Atlas (TCGA) for the target regions using Wanderer [32] and found *ANK2* to be hypermethylated at three of the four CpG probes in this region (cg 25915539, cg17665652, and cg08448479), which corresponds to the hypermethylation that we observed in the orthologous region in dog (Figure 5A). Interestingly, even though the amount of CpGs in this region is much less in humans than dogs, these three hypermethylated human CpGs are all conserved between the two species and were shown to be hypermethylated in the CMT dog samples (Figure 2A). The expression data from TCGA also indicated a downregulation of *ANK2* in HBC, which matches our canine data (Figure 5B), and these data are further supported by the survival plot of *ANK2* that shows a lower survival rate with decreased *ANK2* expression (Figure 5C). The human *EPAS1* region orthologous to the hypermethylated canine *EPAS1* region that we analyzed contained 2 hypermethylated CpG probes according to the TCGA data (Figure 5D). Furthermore, the *EPAS1* expression level was shown to be downregulated in HBC and the survival rate was decreased in accordance with *EPAS1* downregulation (Figure 5E,F).

To evaluate whether these trends would hold for clinical cfDNA samples, we isolated some from human plasma, obtained from both normal and HBC patients, and subjected it to bisulfite PCR sequencing. Unexpectedly, the *EPAS1* target region did not show any methylation in this region for either HBC (four samples) or normal (two samples consisting of five and six pooled cfDNA isolates, respectively) [33], but this does coincide with the unexpected lack of cfDNA hypermethylation in CMT (Figure 4B). The hypermethylation trends shown in *EPAS1* CMT tissue and HBC data from TCGA seems to not hold up in either canine or human cfDNA. On the other hand, sequencing results of the *ANK2* target region from human cfDNA was more in line with what was suggested by the TCGA data. We sequenced colonies from four separate samples of HBC cfDNA and from 5 pooled normal cfDNA samples. Of the three CpGs indicated to be hypermethylated in this region from the TCGA data, only cg08448479, designated as CpG 5, was also shown to be more methylated in HBC cfDNA according to our sequencing results (Figure 5G). CpG 5 showed an increase in methylation from 71.88% in normal to 85.48% in HBC, which is consistent with the largest increase in methylation seen for the orthologous canine CpGs from 31.04% in normal to 62.07% in CMT (Figure 5G). We attempted to perform the qMSP procedure on pooled samples of HBC cfDNA (14 samples) and pooled sample of normal human cfDNA (5 samples) with MSP primers that incorporated CpGs 1, 2, and 3 and CpGs 5 and 6 in the forward and reverse primers, respectively, but no significant difference was observed (Appendix A). In both the TCGA dataset and the cfDNA sequencing, the *ANK2* target indicated hypermethylation, even if limited to only a few CpGs, which is concurrent with what was seen in the canine CMT data.

## 3. Discussion

CMT is a common malignancy in female dogs and, moreover, in the scope of using dogs as animal models for human disease, its study can be translated to HBC. The field of comparative oncology using canine models have been increasing in popularity, yet the epigenome of canines with CMT remains incomplete. Methylation of DNA has been highlighted as a biomarker in both tissue and liquid biopsies [22,23,24], and thus we set out in this study to evaluate potential methylation biomarkers in CMT. This study identified novel regions in the 21st and 1st introns of *ANK2* and *EPAS1*, respectively, that were significantly hypermethylated in CMT tissue and, in the case of *ANK2*, also in cfDNA isolated from dogs suffering from the disease. The *ANK2* region was furthermore shown to be hypermethylated in human cfDNA at a corresponding CpG on the orthologous human region.

*ANK2* encodes multiple AnkyrinB isoforms, which are integral membrane proteins that connect the lipid bilayer to the membrane skeleton [34]. In this present study, it was indicated that *ANK2* mRNA expression was downregulated in CMT and also in HBC based on TCGA data. It is of note that other members of the ankyrin family, namely *ANK1* and *ANK3*, have been shown to be overexpressed in multiple human cancers including breast cancer [35,36], while *ANK2* overexpression has been shown in pancreatic cancer where it contributed to the malignant phenotype [37]. However, other studies found that *ANK2* gene expression was downregulated in colorectal cancer [38] and thyroid cancer [39], which corresponds to what we found in this study. The expression status and role of *ANK2* seems to vary amongst cancer types, and further investigation in a larger variety of cancers is required to better understand its relationship to cancer. Moreover, the role of cancer associated aberrant methylation in *ANK2* remains largely unknown. One previous study compiled datasets for nasopharyngeal cancer and found *ANK2* to be hypomethylated and upregulated, highlighting it as a potential biomarker for this cancer type [40]. Our study found a hypermethylated intron 21 region of *ANK2*, in both tissue and cfDNA, and it was highlighted as a potential tissue and liquid biopsy biomarker for CMT. Yet, the impact of this specific methylation in CMT as well as the fuller extent of *ANK2* methylation, on the promotor region for instance, remains unknown and should be studied further. Furthermore, the orthologous human region does contain hypermethylated CpGs but outside of a CpG island which makes the qMSP method difficult, even more so when considering that of the six CpGs in the region only one seemed to be differentially methylated based on the sequencing data in Figure 5G. The subsequent failure of the qMSP in distinguishing significantly between the HBC and Normal *ANK2* is probably due to there only being one CpG (CpG 5) that is differentially methylated, and the technique is reliant on more than one CpG being differentially methylated. Additional sequencing of this region in more samples would be the best way to highlight the differential methylation in the respective CpGs. Other regions on *ANK2* that contain conserved CpG islands could potentially be more successfully analyzed via the qMSP method and could potentially prove more influential in the downregulation of expression seen in Figure 5B. Pyrosequencing could be another alternative to pursue, as opposed to cloning and Sanger sequencing. We did not observe any methylation difference on the promoter region in our previous MBD-seq data [26]. This orthologous human region should be further studied in the future in tissue and cell culture to further investigate the nature and role of these hypermethylated CpGs play a significant role in HBC. To our knowledge, no other study yet been able to elucidate the state of *ANK2* in either HBC or CMT, which makes our novel investigation of *ANK2* in these diseases very insightful.

*EPAS1*, on the other hand, is relatively well studied in its relation to cancer, including breast cancer. EPAS1 is part of the hypoxia inducible factor (HIF) family and plays an important role in the hypoxic response pathway which is often associated with tumorigenesis [41]. EPAS1 acts as a transcription factor in hypoxic conditions with many of its targets being known to contribute to carcinogenesis, such as VEGF and VEGF-R1 [41,42]. It has been highlighted as an oncogene in numerous cancer types [43,44]. However, EPAS1 has also been described as tumor suppressive in various other cancer types including neuroblastoma, colon cancer, and hepatocellular carcinoma, and shown to reduce tumor growth and improve patient outcomes when overexpressed [45,46,47,48]. In breast cancer, the role and expression levels of EPAS1 have been indicated to be subtype specific. Klahan et al. [49] found EPAS1 to be a subtype specific lymphovascular invasion marker in ER- and HER2+ breast cancer. It has additionally been demonstrated that there is an estrogen dependent downregulation of EPAS1 in ER+ but not ER- HBC and proposed EPAS1 as a negative prognostic marker in HER2+ invasive breast cancer, since overexpression of EPAS1 worsen the prognosis in HER2 and ER+ [50,51].

This present study indicated an overall downregulation of *EPAS1* in CMT and in invasive HBC, yet the role of DNA methylation still remains unclear as, despite the indication of hypermethylation of the proposed region in the first intron of human *EPAS1* shown in the TCGA data, the cfDNA analysis showed no methylation difference between normal and HBC patients. Similarly, the canine cfDNA data did not show any increased methylation in this region. However, significant hypermethylation was noted in CMT tissue compared to paired adjacent normal tissue. This discrepancy between the tissue methylation and cfDNA methylation could potentially be explained by a lesser abundance of circulating tumor DNA (ctDNA) compared to normal cfDNA isolated from the plasma samples used, seeing as even though tumors usually secrete a larger amount of cfDNA in the plasma there is currently no ideal method for separating ctDNA from total cfDNA. ctDNA is furthermore highly fragmented which could hinder its detection [23]. Analyzing the methylation of the *EPAS1* target region in circulating tumor cells (CTCs) could prove insightful. The hypermethylation and mRNA downregulation noted in the CMT tissue is however concurrent with previous studies where *EPAS1* has been shown to be hypermethylated in its promotor with downregulated mRNA expression in non-small cell lung cancer [52] and colon cancer [47]. Although a separate study showed *EPAS1* to be hypomethylated in colon cancer [53]. In CMT tissue, our study of *EPAS1* and its indications of hypermethylation in cancer as opposed to normal remains a novel and intriguing finding and highlights this target as a candidate tissue biomarker for CMT. Yet, future studies should endeavor to delve deeper into the methylation of the targeted intron region in HBC by analyzing tissue and cell culture samples. Elucidating the functional role of this hypermethylation in CMT and potentially HBC, specifically in ER+ subtypes that have previously indicated downregulation in HBC [50] and in HER2 subtypes, where it has been implicated as a prognostic marker [51], is important to understanding the regulation of *EPAS1* and its potential role in CMT and HBC.

Overall, this study set out to investigate CMT tissues and cfDNA isolated from plasma obtained from dogs with CMT for potential biomarkers and attempt to translate the findings to HBC by ways of comparative oncology. This study identified novel hypermethylated intron regions for *ANK2* and *EPAS1* that could potentially serve as tissue biomarkers for CMT. Furthermore, the *ANK2* target also showed hypermethylation in cfDNA isolated from CMT, which highlights it as a potential liquid biopsy biomarker for CMT. In HBC, one CpG from the orthologous human *ANK2* region likewise indicated a hypermethylated trend in cfDNA isolated from HBC patients, which emphasizes the link between CMT and HBC epigenetic regulation.

## 4. Materials and Methods

### 4.1. Ethics

This study, including the materials and methods, was reviewed and approved by the Seoul National University Institutional Review Board/Institutional Animal Care and Use Committee (IACUC# SNU-170602-1, 02 June 2017/IRB#SNU 16-10-063, 6 October 2016), Samsung Medical Center IRB (#SMC2016-07-129-015, 12 July 2016), and the Seoul Metropolitan Government—Seoul National University Boramae Medical Center IRB (#20161123/16-2016-99/121, 23 November 2016).

### 4.2. Tissue and Plasma Samples

Tissue samples used in this study consisted of surgically removed CMT tissue and paired adjacent normal tissue obtained from a variety of dog breeds. Tumors identified as carcinomas were selected, yet the selection was not subtype specific (simple, complex, and ductal carcinomas) seeing as the MBD-seq data (accession number PRJNA601533) [26] did not show any subtype specific differences in methylation patterns for our targets. Canine blood samples were taken from dogs diagnosed with mammary carcinoma and normal samples were obtained from healthy dogs without cancer. A detailed roster of the dogs used can be found in Appendix A. Depending on the dog size, 2–4 mL of blood was collected in Vacuette EDTA tubes (Greiner Bio-One, Kremsmunster, Austria).

Human blood samples were obtained from Samsung Medical Center, Seoul. After written informed consent was given, 4–6 mL of human blood was obtained from HBC patients scheduled to undergo surgery and from healthy control subjects that indicated no breast abnormalities upon examination. A detailed roster of the human subjects can be found in Appendix A.

Both canine and human plasma samples were separated from the whole blood immediately after blood collection. An equal volume of Ficoll-Paque PLUS (GE Healthcare, Orsay, France) was added to each blood sample and centrifuged for 30 min, at 500× *g*, 18 °C without brake. Plasma was collected from the supernatant and stored at −80 °C until use.

### 4.3. Correlation Analysis between Methylation and Gene Expression

We performed an integrative analysis of MBD-seq (accession number PRJNA601533) [26] and RNA-seq data (SRA accession number: SRR8741587-SRR8741602) [25] from 8 overlapped tissue samples to identify canine mammary gland DNA methylation markers. We first selected DMRs which are located in a CpG island on the gene body and examined the correlation between DNA methylation and gene expression for each sample. To inspect the impact of DNA methylation on the local regulation of gene expression, the Pearson correlation (r) was calculated between the read count for DMRs located in CpG regions and the expression values of the corresponding genes. Log (fpkm+1) values were used to avoid taking log0 in the case of there being 0 counts. |r| > 0.3 and an adjusted *p*-value < 0.05 were set as the cutoffs for a significant correlation.

### 4.4. DNA Isolation

Genomic DNA was isolated from 25 mg samples of paired CMT and adjacent normal tissues using the DNEasy Blood & Tissue kit (Qiagen, Hilden, Germany) according to the manufacturer’s protocol. The recovered gDNA was quantified using a nanodrop spectrophotometer. Circulating cfDNA was isolated from both human (Normal and Breast Cancer patients) and canine (Normal and CMT patients) plasma samples using the QIAamp Circulating Nucleic Acid Kit (Qiagen, Hilden, Germany) according to the manufacturer’s instructions. In short, 500 uL of plasma was brought up to 1 mL with PBS and lysed with proteinase K and Buffer ACL before being bound, washed, and eluted from QIAamp mini columns on a vacuum manifold. Recovered cfDNA was quantified with a Qubit 3.0 Fluorometer using the Qubit HS dsDNA Assay (Invitrogen, Carlsbad, CA, USA), according to the manufacturer’s protocol.

### 4.5. Bisulfite Sequencing

DNA samples were Bisulfite treated using the EZ DNA Methylation-Lightning kit (Zymo Research, Irvine, CA, USA), according to the manufacturer’s instructions. Bisulfite treated DNA was quantified using the Qubit ssDNA Assay on the Qubit 3.0 Fluorometer (Invitrogen, Carlsbad, CA, USA). Paired samples were brought to equal concentrations by diluting with water as appropriate before qMSP was performed.

Randomly selected samples of Bisulfite treated gDNA was sequenced to ascertain what the general methylation patterns are for both the *ANK2* and *EPAS1* targets in both CMT and normal tissue. Bisulfite Sequencing PCR (BSP) was conducted using the BSP primers (Appendix A) and HotStart Taq DNA Polymerase (Bioneer, Daejeon, Korea). Successful amplification was validated via gel electrophoresis on 2.0% Agarose gels. PCR amplicons were purified using the MEGAquick-spin Plus Total Fragment DNA Purification kit (Intron Biotechnology, Seongnam, Korea) according to the manufacturer’s protocol. The amplicons were then ligated into pGEM T-Easy vector (Promega, Madison, WI, USA) and transformed into competent E. coli cells via heat-shock at 45 °C for 45 s and plated on LB Agar plates that had been treated with Ampicillin, X-Gal, and IPTG to allow for Blue/White screening of colonies. White colonies were picked and sent for Sanger sequencing (Macrogen, Seoul, Korea). cfDNA from human samples, HBC and normal, were similarly subjected to BSP sequencing.

### 4.6. Quantitative Methylation-Specific PCR (qMSP)

To investigate the methylation of both cancer and normal gDNA and cfDNA, quantitative MSP was performed on each sample with MSP primers (Appendix A) using the CFX96 Real-time PCR Detection System (Bio-Rad Laboratories, Hercules, CA, USA). The same samples were also subjected to quantitative PCR using the BSP primers (Appendix A), which were designed to flank the region of interest, to normalize the MSP readings and allow for the calculation of the Methylation Index, which was based on the demethylation index first introduced by Akirav et al. [28]. Each qMSP reaction contained 0.5 units Hotstart Taq DNA Polymerase (Bioneer, Daejeon, Korea), 0.625 mM MgCl2, 0.2 mM dNTPs, 0.5X SYBER Green (Life Technologies, Carlsbad, CA, USA) and 10 pmol each of forward and reverse primers in a 20 uL total volume. For the gDNA samples, 5–30 ng of template DNA was added, whereas 120–1200 pg was added for the cfDNA samples. Each sample was run in triplicate using both the MSP and BSP primer sets and the thermal cycler conditions were as follows: 95 °C for 15 min, denaturation at 95 °C for 30 s, annealing at Annealing Temperature (Appendix A) for 30 s, elongation at 72 °C for 30 s and final elongation at 72 °C for 5 min.

All primers were manually designed and checked using the OligoEvaluator™ webtool. MSP primers were designed with at least 6 CpGs included in the set of primers, and with at least one CpG located in the last 3 bases at the 3′ ends of the primer. BSP primers were designed with at least 4 non-CpG C’s included in the primer set. For canine primer optimization, there are not any universally methylated and unmethylated controls commercially available. In-house controls were thus made. In short, fully methylated canine controls were made by treating ~1 µg of canine gDNA with MssI methyltransferase (New England Biolabs, Ipswich, MA, USA) according to the manufacturer’s protocol. Fully unmethylated DNA was made by performing PCR with primers that flank the region of interest. The resulting amplimers were thus devoid of any methylation and served as completely unmethylated control template. Human primers were optimized on fully methylated and fully unmethylated human HCT116 DKO DNA (Zymo Research, Irvine, CA, USA).

### 4.7. Human TCGA Data

Using liftOver [54], we were able to convert the canine (CanFam3.1) genome coordinates of the differentially methylated intron regions of *ANK2* and *EPAS1* that we identified from the MBD-seq data to their orthologous regions on the human genome (Hg38). BLAST (https://blast.ncbi.nlm.nih.gov) was used to align the nucleotide sequences and determine the amount of sequence homology.

HBC data for the *ANK2* and *EPAS1* intron targets was obtained using Wanderer [32], which utilizes 450k Infinium Chip methylation arrays and Illumina HISeq RNA-seq data from TCGA. Survival plots were generated using Kaplan–Meier Plotter with the auto selected cutoff enabled, which utilizes the expression data for the genes of interest, *ANK2* (202920_at) and *EPAS1* (200878_at), and the relapse free survival data of 3951 patients obtained from the GEO database [55]. The Kaplan–Meier Plotter software groups were expressed as high or low according to the median expression and then a Kaplan–Meier plot was used to compare the two groups [55].

## Figures and Tables

**Figure 1 ijms-21-08697-f001:**
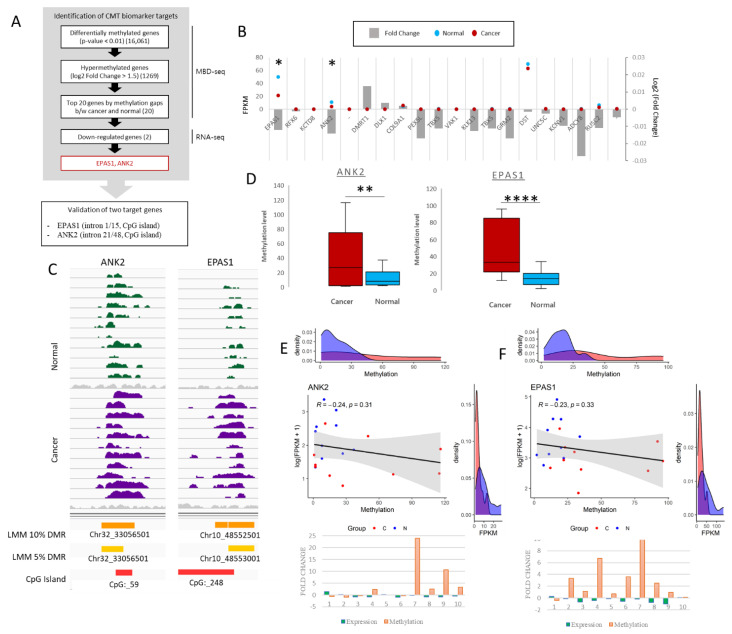
Selection of *ANK2* and *EPAS1* as canine mammary tumor (CMT) hypermethylated targets. **(A**) Schematic overview of target selection using MBD-seq and RNA-seq. Each step indicates the number of identified genes at the end of that step. (**B**) RNA-seq gene expression in FPKM of top 20 hypermethylated genes; blue dots = normal, red dots = CMT, * = *p*-value < 0.05. Fold change (log2) for each target gene indicated with grey bar graph. (**C**) IGV peak calling for *ANK2* and *EPAS1* in both normal and cancer. Differential methylation assigned via LMM with both 5% and 10% threshold, and CpG island presence is indicated. (**D**) Overall methylation levels of *ANK2* and *EPAS1* from MBD-seq data for 11 paired CMT and adjacent normal samples. EdgeR. ** = *p*-value < 0.01, **** = *p*-value < 0.0001. (**E**) *ANK2* and (**F**) *EPAS1* correlation plots between expression (FPKM) and methylation of 10 paired CMT (red) and normal (blue) samples with matching density plots for both FPKM and methylation. Pearson correlation |r| value and *p* value indicated. Fold change graphs are also depicted for 10 CMT samples indicating expression (green) and methylation (orange).

**Figure 2 ijms-21-08697-f002:**
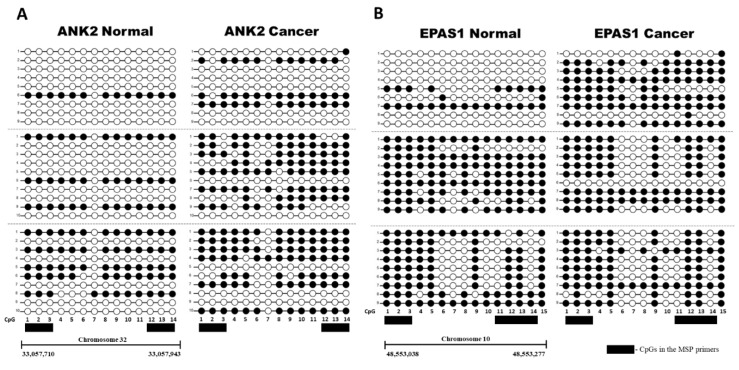
Bisulfite Sequencing PCR (BSP) of *ANK2* and *EPAS1*. Between 9 and 10 colonies were sequenced for three paired normal and CMT samples (separated by grey dotted line) for (**A**) *ANK2* and (**B**) *EPAS1*. Methylation of a particular CpG is indicated by a black circle, and non-methylation is indicated by a white circle. The CpGs that were included in the forward and reverse MSP primers for each respective target with a black box.

**Figure 3 ijms-21-08697-f003:**
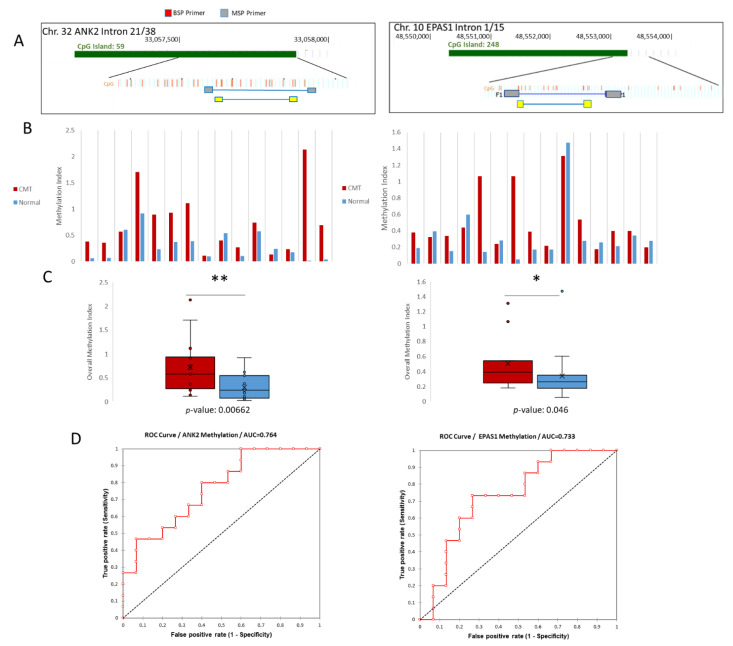
Quantitative Methylation Specific PCR (qMSP) validation of *ANK2* and *EPAS1* hypermethylation. (**A**) Schemes for both *ANK2* and *EPAS1* indicating the target region in terms of its position on the CpG island of its respective intron. BSP primers (grey) are shown to flank the regions targeted by the MSP primers (yellow). (**B**) Methylation Indexes for both *ANK2* and *EPAS1*. CMT samples are shown in red and paired normal samples are shown in blue. (**C**) Overall methylation indexes for both *ANK2* and *EPAS1* are shown. CMT in red, normal in blue. * = *p*-value < 0.05, ** = *p*-value < 0.01. (**D**) ROC curve analyses for *ANK2* and *EPAS1*.

**Figure 4 ijms-21-08697-f004:**
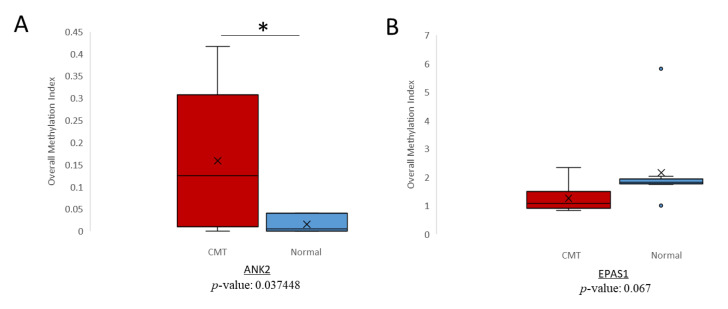
Canine Mammary Tumor (CMT) hypermethylation analysis in cfDNA. Overall methylation index for (**A**) *ANK2* and (**B**) *EPAS1* in CMT (red) and normal (blue) cfDNA. * = *p*-value < 0.05.

**Figure 5 ijms-21-08697-f005:**
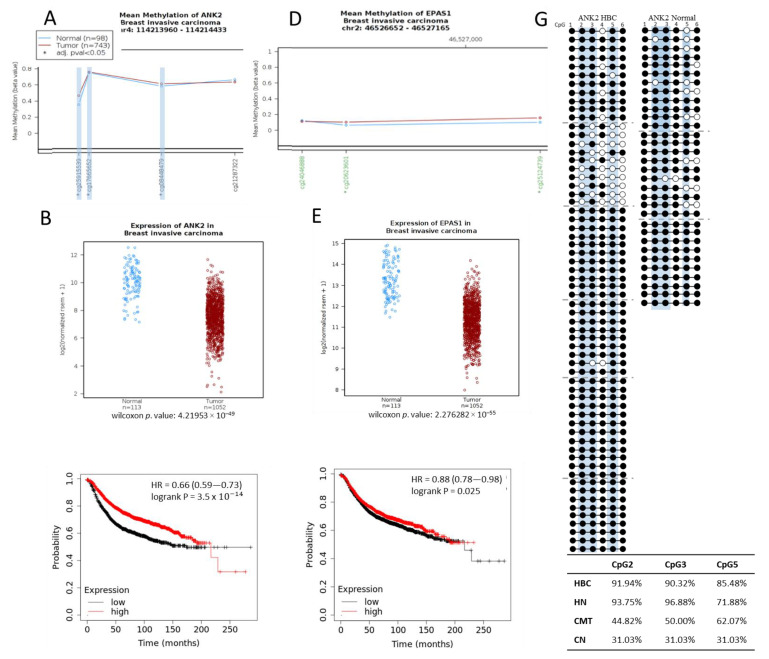
Hypermethylation of orthologous human target *ANK2* and *EPAS1* regions. Mean Methylation (beta value) plots for orthologous human (**A**) *ANK2* and (**D**) *EPAS1* regions for normal (blue) and invasive breast cancer patient(red) data obtained from TCGA. Specific methylation probes, annotated at the bottom of the graph are indicated to be in a CpG island when written in green (only seen for *EPAS1*), and designated as significant (*) with an adjusted *p*-value < 0.05. Three significantly hypermethylated CpGs from *ANK2* are highlighted in light blue to indicate that they are conserved in the dog genome. (**B**) *ANK2* and (**E**) *EPAS1* log2 (normalized rsem +1) expression data from TCGA for normal (blue) and invasive breast cancer patients (red). Wilcoxon *p*-values are indicated. (**C**) *ANK2* and (**F**) *EPAS1* Kaplan-Meier plots indicting relapse-free survival with high and low expressions each. (**G**) *ANK2* sequencing results for cfDNA from normal and HBC patients. Between 10 and 11 colonies were sequenced for 6 individual HBC cfDNA samples (separated by grey dotted lines) and between 10 and 12 colonies were sequenced from 3 separate normal cfDNA samples. Methylation of a specific CpG is indicated by a black circle and non-methylation by a white circle. The CpGs at positions 2, 3, and 5 are highlighted in light blue as these are the CpGs that correlate with the hypermethylated CpG probes indicated in (**A**) and the respective percentage methylation for these CpGs in the HBC and normal human (HN) cfDNA samples, as well as in canine CMT and normal (CN) tissue (Figure 2A), are indicated in the inserted table.

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
