# Peer review of "ANK2* Hypermethylation in Canine Mammary Tumors and Human Breast Cancer"

_ijms, 2020, doi:10.3390/ijms21228697_

Round 1

Reviewer 1 Report

The manuscript Hypermethylation on ANK2 and EPAS1 as diagnostic markers for both Canine Mammary Tumor and Human Breast Cancer by Schabort et al. describes the characterization of the differential methylation status of genome regions between canine mammary tumors and normal tissues. The obtained genome regions are analyzed and compared in human tumors, too.

The manuscript is interesting and well written. The possibility of having the same biomarkers for canine and human mammary tumors is very interesting.

I have several criticisms

No data about the four human mammary tumors are provided. Which kind of human breast cancer they are? Are they similar by a molecular point of view? They are, in any case, too few to reach a proof.

It should be interest investigate the methylation of DST and RUSC2, too. The four gene signature would be more informative than one single gene. Moreover, for the comparison with human, were ANK2 region does not contain CpGs this is even more important.

The cohort of cfDNA cases must be expanded in both dogs and humans to deepen the hypermethylation trend of the four genes in cfDNA. This data would be the most interesting for a possible future  translation  into the clinic.

Author Response

Response to Reviewer 1

The manuscript Hypermethylation on ANK2 and EPAS1 as diagnostic markers for both Canine Mammary Tumor and Human Breast Cancer by Schabort et al. describes the characterization of the differential methylation status of genome regions between canine mammary tumors and normal tissues. The obtained genome regions are analyzed and compared in human tumors, too.

The manuscript is interesting and well written. The possibility of having the same biomarkers for canine and human mammary tumors is very interesting.

I have several criticisms

No data about the four human mammary tumors are provided. Which kind of human breast cancer they are? Are they similar by a molecular point of view? They are, in any case, too few to reach a proof.

The human breast cancer samples were summarized in Supplementary Table S4 in the revised manuscript. As you can see in the revised manuscript, we conducted qMSP on an increased amount of pooled human cfDNA samples (14 HBC and 5 normal), but there was no significant difference in the methylation (Supplementary Figure S1). The qMSP method was unable to properly distinguish between the HBC and normal probably due to there only being 1 differentially methylated CpG in the region. Additional sequencing of more samples would possibly give the best indication of the differential methylation of the region, and we were able to sequence 3 more HBC samples and 2 more Normal samples, which did increase the hypermethylation trend seen between HBC and Normal for CpG 5 (Fig 5G). Cloning and sequencing even more samples would potentially deepen this trend further but requires extensive time.

It should be interest investigate the methylation of DST and RUSC2, too. The four gene signature would be more informative than one single gene. Moreover, for the comparison with human, were ANK2 region does not contain CpGs this is even more important.

Initially we left out DST and RUSC2 since even though they showed hypermethylation in the MBDseq data and was downregulated in the RNAseq data, this downregulation was not significant. ANK2 and EPAS1 did show significant downregulation and were thus chosen for further study. DST and RUSC2 may very well still be viable as methylation biomarkers and it will be interesting to revisit them in the future, yet to investigate them as well is beyond the scope of this present study.

The cohort of cfDNA cases must be expanded in both dogs and humans to deepen the hypermethylation trend of the four genes in cfDNA. This data would be the most interesting for a possible future  translation  into the clinic.

We agree that additional samples of both canine and human cfDNA would be valuable to this present study. In the revised manuscript, we adjusted our canine ANK2 qMSP protocol and analyzed pooled samples from 19 CMT patients and 10 Healthy subjects. The reworked protocol and the additional samples did indeed deepen the hypermethylation to a significant point (Fig 4A). As for the human cfDNA samples, as mentioned earlier, 3 additional HBC and 2 normal samples were sequenced, which emphasized the hypermethylation trend further (Fig 5G). Additionally HBC and Normal human cfDNA samples were pooled and qMSP was conducted but was unable to show significant hypermethylation (Supplementary Fig S1) probably due to the Ank2 region not being in a CpG island in Human and the qMSP procedure not being suitable to detect differential methylation in only 1 CpG (CpG5). More samples would have to be sequenced, either through Sanger sequencing as was done thus far or Pyrosequencing can also be considered. Sequencing of additional human samples would just require additional time.

Reviewer 2 Report

This paper describes an interesting contribution in the field of comparative oncology, with a potential impact for canine mammary cancer. However, there are some important points that should be discussed in order to improve the comprehensibility of the present manuscript.

Title

Please change the title “Hypermethylation on ANK2 and EPAS1 as diagnostic markers for both Canine Mammary Tumor and Human Breast Cancer” as it does not accurately represent the present study. These results showed a significant difference in methylation between CMT samples and canine “normal” mammary samples, but do not support ANK2 and EPAS1 hypermethylation as diagnostic markers.

Abstract

Line 11: Please change “Canine Mammary Tumor (CMT) is one of the most common tumor types found in dogs” to “Canine Mammary Tumors (CMT) constitute the most common tumor types found in female dogs”.

Line 15: Please change “MBD-seq” to “Methyl-binding domain sequencing (MBD-seq)”

Line 18: Please change “cfDNA” to “cell free DNA (cfDNA)”

Lines 20-21: Please change “plasma cell free DNA (cfDNA) of CMT” to “cfDNA) of CMT”.

Introduction

Line 27: Please change “Canine Mammary Tumor (CMT) is the most” to “Canine Mammary Tumors (CMT) are the most”

Line 40: The sentence “tumor suppressor and oncogenes, respectively, are observed during cancer” is not clear.

Lines 44-46: With regard to the sentence: “Biomarkers from tissue still remain the standard for cancer diagnosis and prognosis, yet in recent years the interest in liquid biopsy biomarkers have risen” please provide the references.

Line 57: Please change “CMT tissue” to “CMT tissues”.

Results

Some methods were performed in a distinct number of samples, namely for QMSP validation of ANK2 and EPAS1 hypermethylation (10 and 15 samples, respectively) (Figure 3). Please explain these differences.

Lines 142-143: The sentence “To evaluate the methylation of the target regions as potential liquid biopsy biomarkers in both CMT” is not clear.

Lines 143-144: The sentence “we conducted qMSP with quantitative BSP for normalization on cfDNA that was isolated from CMT patient and normal canine plasma samples” is not clear. Please correct to “we conducted qMSP with quantitative BSP for normalization on cfDNA isolated from plasma samples of CMT and normal female dogs”.

Lines 189-192: Please correct “Fig. 4E” to “Fig. 5E”.

Discussion:

Line 213: Please change “in dogs” to “in female dogs”.

Line 220: Please change “from dogs and suffering from the disease” to “from dogs suffering from the disease”.

Line 271: Please change “hypermethylation and downregulation” to “hypermethylation and mRNA downregulation”.

Line 273: Please change “Non-small” to “non-small”.

Line 280: Please change “were it has been implicated as prognostic markers” to “where it has been implicated as a prognostic marker”.

Line 281: Please change “its potential tumor suppressive role in CMT and HBC” to “its potential role in CMT and HBC”. According to the studies pointed by authors, EPAS1 role depends on the tumor subtype, thus it is not correct to assume a general “tumor suppressive role”.

Line 282-283: Please change “set out to investigate CMT tissue and plasma for potential biomarkers and attempt to translate the findings to HBC” to “set out to investigate CMT tissues and cfDNA isolated from plasma obtained from dogs with CMT for potential biomarkers and attempt to translate the findings to HBC”.

Material and methods

A subsection regarding tissue samples should be provided, including the number of canine samples for each methodology, as well as information on the dog patients and histological tumor types.

Similarly, there is a lack of information regarding the source of the human plasma samples, which seam a rather low number of samples. Please comment.

Please provide the inclusion criteria for animal/tumor selection.

How did the authors define “normal” canine mammary tissue?

Did the authors evaluate the protein expression of ANK2 and EPAS1 in the canine tissue samples?

Author Response

Response to Reviewer 2

This paper describes an interesting contribution in the field of comparative oncology, with a potential impact for canine mammary cancer. However, there are some important points that should be discussed in order to improve the comprehensibility of the present manuscript.

Title

Please change the title “Hypermethylation on ANK2 and EPAS1 as diagnostic markers for both Canine Mammary Tumor and Human Breast Cancer” as it does not accurately represent the present study. These results showed a significant difference in methylation between CMT samples and canine “normal” mammary samples, but do not support ANK2 and EPAS1 hypermethylation as diagnostic markers.

 We changed our title to more accurately reflect the findings of the study as the reviewer suggested.

Abstract

Line 11: Please change “Canine Mammary Tumor (CMT) is one of the most common tumor types found in dogs” to “Canine Mammary Tumors (CMT) constitute the most common tumor types found in female dogs”.

This was changed as the reviewer suggested. Line 15

Line 15: Please change “MBD-seq” to “Methyl-binding domain sequencing (MBD-seq)”

This was changed as indicated. Line 20

Line 18: Please change “cfDNA” to “cell free DNA (cfDNA)”

This was changed as indicated. Line 23

Lines 20-21: Please change “plasma cell free DNA (cfDNA) of CMT” to “cfDNA) of CMT”.

 This was changed as indicated. Line 25

Introduction

Line 27: Please change “Canine Mammary Tumor (CMT) is the most” to “Canine Mammary Tumors (CMT) are the most”

This was changed as indicated. Line 34

Line 40: The sentence “tumor suppressor and oncogenes, respectively, are observed during cancer” is not clear.

This sentence was reworded to make it clear. Line 47

Lines 44-46: With regard to the sentence: “Biomarkers from tissue still remain the standard for cancer diagnosis and prognosis, yet in recent years the interest in liquid biopsy biomarkers have risen” please provide the references.

Reference was added as indicated. Line 53

Line 57: Please change “CMT tissue” to “CMT tissues”.

 This was changed as indicated. Line 64

Results

Some methods were performed in a distinct number of samples, namely for QMSP validation of ANK2 and EPAS1 hypermethylation (10 and 15 samples, respectively) (Figure 3). Please explain these differences.

Additional ANK2 tissue samples were analyzed in the revised manuscript to reflect the number of samples used in the EPAS1 cohort (15 samples). Figure 3 was adjusted accordingly.

Lines 142-143: The sentence “To evaluate the methylation of the target regions as potential liquid biopsy biomarkers in both CMT” is not clear.

This sentence was reworded to make it clear. Line 153-156

Lines 143-144: The sentence “we conducted qMSP with quantitative BSP for normalization on cfDNA that was isolated from CMT patient and normal canine plasma samples” is not clear. Please correct to “we conducted qMSP with quantitative BSP for normalization on cfDNA isolated from plasma samples of CMT and normal female dogs”.

This was corrected. Line 153-156

Lines 189-192: Please correct “Fig. 4E” to “Fig. 5E”.

 This was corrected as indicated. Line 208

Discussion:

Line 213: Please change “in dogs” to “in female dogs”.

This was changed as indicated. Line 243

Line 220: Please change “from dogs and suffering from the disease” to “from dogs suffering from the disease”.

This was changed as indicated. Line 250

Line 271: Please change “hypermethylation and downregulation” to “hypermethylation and mRNA downregulation”.

This was changed as indicated. Line 307

Line 273: Please change “Non-small” to “non-small”.

This was changed ass indicated. Line 309

Line 280: Please change “were it has been implicated as prognostic markers” to “where it has been implicated as a prognostic marker”.

This was changed as indicated. Line 316-317

Line 281: Please change “its potential tumor suppressive role in CMT and HBC” to “its potential role in CMT and HBC”. According to the studies pointed by authors, EPAS1 role depends on the tumor subtype, thus it is not correct to assume a general “tumor suppressive role”.

The reviewer is correct in indicating the faulty assumption that we made. This was changed accordingly. Line 318

Line 282-283: Please change “set out to investigate CMT tissue and plasma for potential biomarkers and attempt to translate the findings to HBC” to “set out to investigate CMT tissues and cfDNA isolated from plasma obtained from dogs with CMT for potential biomarkers and attempt to translate the findings to HBC”.

This was changed as indicated. Line 319-320

Material and methods

A subsection regarding tissue samples should be provided, including the number of canine samples for each methodology, as well as information on the dog patients and histological tumor types.

Similarly, there is a lack of information regarding the source of the human plasma samples, which seam a rather low number of samples. Please comment.

 A subsection on the tissue and plasma samples were added to the methods section. Additionally, Supplementary Tables S3 and S4 were added to summarize the information on the dog and human samples.

As for the low number of human samples, in the revised manuscript we additionally conducted qMSP on pooled samples of HBC (14 samples) and normal (5 samples) cfDNA, but the qMSP method was unable to significantly differentiate between HBC and normal cfDNA methylation levels (Supplementary Fig S1). This is likely due to the human region not being in a CpG island and the differential methylation seems to be mostly just on 1 CpG (CpG 5) and the qMSP method requires more CpGs to be differentially methylated to properly distinguish between samples. Sequencing thus seems to be the best route to investigate this region. We did sequence 3 additional HBC and 2 normal samples which did deepen the hypermethylation trend noticed in the few samples already sequenced (Fig 5G). Even more samples of HBC and normal would probably deepen the trend even more, however, the procedure of sequencing takes more time than we had.

Round 2

Reviewer 1 Report

The revised version of the manuscript is now acceptable for publication on IJMS

Author Response

 We would like to thank the reviewer for their review.

Reviewer 2 Report

This manuscript shows improvements but there are still some important points that need further attention.

Please change the title “Hypermethylation on ANK2 in both Canine Mammary Tumor and Human Breast Cancer” to “ANK2 Hypermethylation in Canine Mammary Tumors and Human Breast Cancer”.

Lines 33-35: Please include reference(s) in the sentence: “Apart from the clear veterinary benefit to understanding the nature of this cancer more clearly, studying CMT can also be beneficial in the field of comparative oncology.”

Line 36 and 298: Please change “model animals” to “animal models”.

Lines 40-42: Please include reference(s) in the sentence: “The similar environments to humans that dogs often inhabit as companion animals are particularly interesting from an epigenetic standpoint, as environmental stimuli have been known to influence epigenetics.”

Lines 53-55: Please correct the sentence: “Aberrant global hypomethylation and localized, specific hypermethylation of tumor suppressors and hypomethylation of oncogenes are observed during cancer”.

Line 58: Please change “Biomarkers from tissue” to “Tissue biomarkers”.

Fig. 3B- Methylation index. Please correct the labels of the y axis on the left, to CMT and Normal.

Line 159: Please correct “disease” to “disease.”

Lines 252-259: The sentences: “The failure of the qMSP in distinguishingsignificantly between the HBC and Normal ANK2 is probably due to there only being 1 CpG (CpG 5) that is differentially methylated, and the technique is reliant on more than 1 CpG being differentially methylated. Additional sequencing of this region in more samples would be the best way to highlight the differential methylation in the respective CpGs. Other regions on ANK2 that contain conserved CpG islands could potentially be more successfully analyzed via the qMSP method and could potentially prove more influential in the downregulation of expression seen in Fig 5B.” should be relocated to the Discussion section.

Line 315: Please change “cancer” to “cancer types”.

In Material and methods, the subsection on Tissue and plasma samples should appear after 4.1 Ethics.

Lines 425: With regard to Supplementary Table S3, it should be more detailed, in a similar way to Supplementary Table S4, but including the following columns: ID number (1,2,3,…), age, breed, histological type (simple, complex, others).

Figure S1 should be mentioned in the text.

Lines 425: Please rename “Supplementary Table S3” to “Supplementary Table S2”.

Lines 431: Please rename “Supplementary Table S4” to “Supplementary Table S3”.

Lines 471 and 482: Please rename “Supplementary Table S2” as “Supplementary Table S4”

Please provide the methodology regarding the survival analysis depicted in Figure 5 C and F. In addition, please clarify how was ANK2 and EPAS1 expression evaluated, as well as the criteria to separate high vs. low expression.

The authors did not answer to the following question in the previous review: was the protein expression of ANK2 and EPAS1 evaluated in the canine tissue samples?

Author Response

Response to Reviewer 2

This manuscript shows improvements but there are still some important points that need further attention.

Please change the title “Hypermethylation on ANK2 in both Canine Mammary Tumor and Human Breast Cancer” to “ANK2 Hypermethylation in Canine Mammary Tumors and Human Breast Cancer”.

We Changed the title as the reviewer suggested.

Lines 33-35: Please include reference(s) in the sentence: “Apart from the clear veterinary benefit to understanding the nature of this cancer more clearly, studying CMT can also be beneficial in the field of comparative oncology.”

References were added (Line 35).

Line 36 and 298: Please change “model animals” to “animal models”.

This was changed accordingly (Line 36 & 234).

Lines 40-42: Please include reference(s) in the sentence: “The similar environments to humans that dogs often inhabit as companion animals are particularly interesting from an epigenetic standpoint, as environmental stimuli have been known to influence epigenetics.”

References were included (Line 42).

Lines 53-55: Please correct the sentence: “Aberrant global hypomethylation and localized, specific hypermethylation of tumor suppressors and hypomethylation of oncogenes are observed during cancer”.

This sentence was corrected (Line 45).

Line 58: Please change “Biomarkers from tissue” to “Tissue biomarkers”.

This was changed accordingly (Line 49).

Fig. 3B- Methylation index. Please correct the labels of the y axis on the left, to CMT and Normal.

This was corrected as the reviewer pointed out.

Line 159: Please correct “disease” to “disease.”

This was corrected (Line 139).

Lines 252-259: The sentences: “The failure of the qMSP in distinguishingsignificantly between the HBC and Normal ANK2 is probably due to there only being 1 CpG (CpG 5) that is differentially methylated, and the technique is reliant on more than 1 CpG being differentially methylated. Additional sequencing of this region in more samples would be the best way to highlight the differential methylation in the respective CpGs. Other regions on ANK2 that contain conserved CpG islands could potentially be more successfully analyzed via the qMSP method and could potentially prove more influential in the downregulation of expression seen in Fig 5B.” should be relocated to the Discussion section.

We agree with the reviewer that this segment is better suited to the discussion section and moved it accordingly (Lines 262-269).

Line 315: Please change “cancer” to “cancer types”.

This was changed accordingly (Line 251).

In Material and methods, the subsection on Tissue and plasma samples should appear after 4.1 Ethics.

The Tissue and Plasma section was moved as the reviewer suggested and the subsection numberings were adjusted accordingly.

Lines 425: With regard to Supplementary Table S3, it should be more detailed, in a similar way to Supplementary Table S4, but including the following columns: ID number (1,2,3,…), age, breed, histological type (simple, complex, others).

Supplementary Table S2 (changed from S3 as suggested by the reviewer in a later comment) was expanded with greater detail per the reviewer’s suggestion.

Figure S1 should be mentioned in the text.

Figure S1 was indeed mentioned in the text in Line 205.

Lines 425: Please rename “Supplementary Table S3” to “Supplementary Table S2”.

This was changed accordingly (Line 340).

Lines 431: Please rename “Supplementary Table S4” to “Supplementary Table S3”.

This was changed accordingly (Line 346).

Lines 471 and 482: Please rename “Supplementary Table S2” as “Supplementary Table S4”

This was changed accordingly (Line 402, 416, & 425).

Please provide the methodology regarding the survival analysis depicted in Figure 5 C and F. In addition, please clarify how was ANK2 and EPAS1 expression evaluated, as well as the criteria to separate high vs. low expression.

The survival analysis methodology and criteria for high and low expression was expanded (Lines 448-452).

The authors did not answer to the following question in the previous review: was the protein expression of ANK2 and EPAS1 evaluated in the canine tissue samples?

We apologize for not answering this question in the previous review. We did not analyze the protein expression of ANK2 and EPAS1 in canine tissue. Even though this would be interesting to look at, for the purpose of this study we were more interested in just the methylation status and the mRNA expression of these genes.

Round 3

Reviewer 2 Report

This manuscript shows major improvements. Please address the comment below.

Supplementary Table S2 (histological type column): adenocarcinoma is no longer used in the classification of canine mammary tumors and it is not considered a histological type; instead, please provide the carcinoma histotype (complex, tubular, papillary or other), as in the other samples, if available.

Author Response

This manuscript shows major improvements. Please address the comment below.

Supplementary Table S2 (histological type column): adenocarcinoma is no longer used in the classification of canine mammary tumors and it is not considered a histological type; instead, please provide the carcinoma histotype (complex, tubular, papillary or other), as in the other samples, if available.

  • Supplementary Table S2 was changed accordingly. We would like to thank the reviewer for their careful and thorough review of this manuscript.